



# Quantifying the trade-offs in re-operating dams for the environment
# in the Lower Volta River
Afua Owusu[a b], Jazmin Zatarain Salazar[b], Marloes Mul[a], Pieter van der Zaag[a c], Jill Slinger[b d]
[a] Land and Water Management Department, IHE Delft Institute for Water Education, Westvest 7, 2611
AX Delft, The Netherlands
[b] Faculty of Technology, Policy and Management, TU Delft, Jaffalaan 5, 2628 BX Delft, The Netherlands
[c] Faculty of Civil Engineering and Geosciences, TU Delft, Stevinweg 1, 2628 CN Delft, The Netherlands
[d] Institute of Water Research, Rhodes University, Drosty Rd, Grahamstown, 6139, South Africa
**Abstract**
The construction of the Akosombo and Kpong dams in the Lower Volta River Basin in Ghana changed
the downstream riverine ecosystem and affected the lives of downstream communities, particularly
those who lost their traditional livelihoods. In contrast to the costs borne by those in the vicinity of
the river, Ghana as a whole, has enjoyed vast economic benefits from the affordable hydropower,
irrigation schemes and lake tourism that developed after construction of the dams. Herein lies the
challenge; there exists a trade-off between water for river ecosystems and related services on the one
hand, and anthropogenic water demands such hydropower or irrigation on the other. In this study, an
Evolutionary Multi-Objective Direct Policy Search (EMODPS) is used to identify the multi-sectorial
trade-offs that exist in the Lower Volta River Basin. Three environmental flows, previously determined
for the Lower Volta are incorporated separately as an environmental objective. The results highlight
the dominance of hydropower production in the Lower Volta, but show that there is room for
providing environmental flows under current climatic and water use conditions if firm energy
requirement from Akosombo Dam reduces by 12% to 38% depending on the environmental flow
regime that is implemented. There is uncertainty in climate change effects on runoff in this region,
however multiple scenarios are investigated. It is found that climate change leading to increased
annual inflows to the Akosombo Dam reduces the trade-off between hydropower and the
environment while climate change resulting in lower inflows provide the opportunity to strategically
provide dry season environmental flows, that is, reduce flows sufficiently to meet low flow
requirements for key ecosystem services such as the clam fishery. This study not only highlights the
challenges in balancing anthropogenic water demands and environmental considerations in managing
existing dams, but also identifies opportunities for compromise in the Lower Volta River
**Keywords:** Environmental flows, Multi-objective evolutionary optimization, Direct policy search, Volta
River, Akosombo Dam
**Corresponding author:** Afua Owusu, IHE Delft Institute for Water Education, Westvest 7, 2611 AX
Delft, Netherlands, email: a.owusu@un-ihe.org



## 1 Background

Freshwater resources are under increasing pressure worldwide (WWF, 2018; He et al., 2019). As global
population and standards of living have gone up, the capacity of many river basins to meet social,
economic and environmental water demands has declined (Best, 2019; Fitzhugh and Richter, 2004;
Postel and Richter, 2003). In the mid-20th century, many dams were built with ambitious goals for
hydropower generation, flood control and irrigation among others and dam construction is seeing a
resurgence in recent years (Grill et al., 2015; Best, 2019). This phenomenon is occurring, despite the
fact that even the economic justification for many existing dams is being called into question (Ansar
et al., 2014; Flyvbjerg and Bester, 2021), the life cycle emissions of some dams is above the median
emissions for fossil fuel plants (Schlömer et al., 2014; Almeida et al., 2019) and the negative social and
environmental impacts of dams on riparian ecosystems and communities have been established for
some time (WCD, 2000; Stone, 2011; Duflo and Pande, 2007; Richter et al., 2010). Proponents of dam
construction argue that in developing regions, particularly in Africa, the large energy deficit (Hafner et
al., 2018) coupled with high inter-annual rainfall variability and the fact that 75% of the population
live in semi-arid or arid regions (Vörösmarty et al., 2005; Smith, 2004), makes multipurpose dams
important infrastructures for energy and food security. Evidently, tools are required for investigating
operation policies for managing and maximising the benefits of dams and the water resources they
control.
An Evolutionary Multi-Objective Direct Policy Search (EMODPS) framework is one such tool (Giuliani
et al., 2016). EMODPS maps the states of a system, in this case, reservoir levels and time of the year,
to actions, the release of water for different water uses (Giuliani et al., 2016; Zatarain Salazar et al.,
2017). They are therefore useful for determining Pareto approximate reservoir operating policies and
thereby assessing the trade-offs between water users in a river basin. The Pareto approximate or non-
dominated set of solutions are the suite of solutions for which increasing the water allocation to one
user leads to a reduction in the benefit to others. EMODPS uses multi-objective evolutionary
algorithms (MOEAs), stochastic search tools to simultaneously find the Pareto approximate set across
multiple objectives (Reed et al., 2013; Matrosov et al., 2015; Zatarain Salazar et al., 2016; Kiptala et
al., 2018; Hurford et al., 2020). The advantage of MOEAs is that they do not require pre-specifying
preferences across objectives, thereby supporting unbiased *a posteriori* decision making (Reed et al.,
2013; Hurford et al., 2014). Furthermore, MOEAs allow for heterogeneous and non-linear problem
formulations with incommensurable objectives and different risk attitudes across objectives.
Accordingly, non-market objectives can be evaluated alongside conventional economic objectives and
this is particularly useful for including environmental flows (e-flows) and ecosystem services for which


monetary valuation is often difficult and contested (Bingham et al., 1995; Costanza et al., 1997, 2014;
Luisetti et al., 2011).
In many of the studies where MOEAs have been applied, the e-flow objective in the simulation
component of the model either meets a minimum flow release (Zatarain Salazar et al., 2017; Kiptala
et al., 2018; Hurford et al., 2020; Gonzalez et al., 2021) or minimizes the deviation of flow from the
natural, unregulated flow regime (Hurford & Harou, 2014). The former objective, minimum flow
releases, fails to thoroughly capture the essence of e-flows which are the "quantity, timing, and quality
of freshwater flows and levels required to sustain aquatic ecosystems" (Brisbane Declaration, 2018).
The latter, the objective of returning fully to the natural flow regime, is an unlikely objective in many
highly modified and utilized river basins (Acreman et al., 2014; Horne et al., 2017). In this study, a
multi-objective analysis of the trade-offs between key water users and the environment in the heavily
modified Lower Volta River Basin in Ghana is carried out. The environmental objectives are designer
e-flows (Acreman et al., 2014) developed for different ecosystem services in the basin. In contrast to
the aim of restoring a river to a near natural state, designer e-flows define and construct parts of the
flow hydrograph of a river to meet certain desired ecological and social outcomes (Acreman et al.,
2014; Horne et al., 2017). Three designer e-flows, defined for the Lower Volta River in previous studies,
are investigated and compared: one to support the Volta clam (*Galatea paradoxa)* (Owusu, Mul,
Strauch et al., 2022) and the other two to support multiple ecosystem services including fisheries,
aquatic weed control, flood recession agriculture and sediment transport (Mul et al., 2017). In
addition, future climatic scenarios are investigated. This study highlights the challenges faced by dam
operators in balancing environmental and anthropogenic water demands for existing dams in heavily
modified and utilized river basins, and simultaneously investigates the room for compromise in the
case of the Lower Volta River. The focus of this paper is on the potential for compromise amongst
water users in the Lower Volta should demand for power generated from the dams in the basin
change. As such, the implications of the trade-off on power delivery, energy prices and carbon
emissions are not investigated.
In the following section, a description of the Lower Volta River Basin is given, followed by the methods
section in which (i) the simulation model for the lower Volta is described, (ii) the multi-objective
evolutionary optimization set up is explained, (iii) the objective functions are formulated, and (iv)
relevant climate-induced effects on discharge are specified. Next is the results section where we
present the trade-off analysis between e-flows and other water uses in the Lower Volta for the current
baseline scenario and possible future scenarios. We conclude with a discussion on the implications of
implementing e-flows in the Lower Volta and draw lessons for other heavily modified river basins.

## 2 Lower Volta River Basin


The Lower Volta River, located in Ghana is one of four sub-basins in the Volta River Basin in West Africa
(Figure 1). It is located furthest downstream, flowing into the Gulf of Guinea and covering an area of
66700 km$^2$, approximately 16% of the Volta Basin. The most important hydraulic infrastructure in the
Lower Volta is the Akosombo Dam, which was built in 1965 for hydropower production with an
installed capacity of 1038 MW (1,020MW Akosombo Hydro Electric Power Plant, 2021). In 1981, a
smaller 160 MW run-of-the-river dam, the Kpong Dam, also began operation downstream. The lake
created by the Akosombo Dam is the largest man-made lake by surface area at about 8500 km$^2$. It has
an average depth of 18.8 m and holds approximately 148 km$^3$ of water at maximum capacity
(1,020MW Akosombo Hydro Electric Power Plant, 2021).

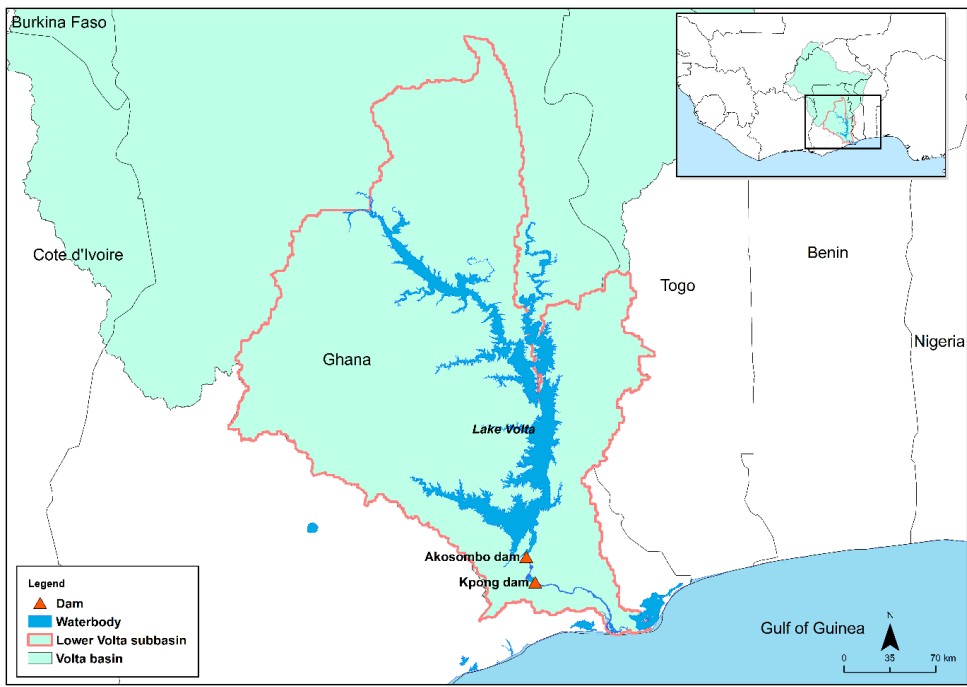

*Figure 1: The Akosombo and Kpong Dams located in the Lower Volta River Basin, which discharges into the Gulf of Guinea*
Construction of the Akosombo Dam led to the resettlement of over 80,000 people (Darko and Tsikata,
2019; Alhassan, 2009) and also changed the dynamic flow regime downstream from one with average
low and high flows of approximately 36 m$^3$/s in March and 5,100 m$^3$/s in September-October
respectively, to a steady flow of about 1,000 m$^3$/s all year round (Ntiamoa-Baidu et al., 2017).
Consequently, the riverine ecosystem changed and so did the lives of downstream communities. Creek
fishing, floodplain agriculture and the clam fishing industries, which together made up three-quarters



of total real income of the Lower Volta riparian population in 1954, collapsed (Moxon, 1969; De-Graft
Johnson, 1999; Tsikata, 2008; Lawson, 1972). In addition, invasive aquatic weeds proliferated,
providing habitat for disease vectors including mosquitoes and snails and thereby increasing the
prevalence of waterborne and water related diseases such as malaria and schistosomiasis (Akpabey
et al., 2017; Gyau-Boakye, 2001). Other environmental costs include changes to the sediment load
leading to erosion along the coastline of Ghana, as well as Togo and Benin (Bollen et al., 2011; Roest,
2018; Appeaning Addo et al., 2020), as well as a reduction in salt water intrusion (Beadle, 1974; People
and Rogoyska, 1969; Nyekodzi et al., 2018). Among the population in the Lower Volta, perceptions of
the Akosombo Dam and the run-of-the-river Kpong Dam downstream, are still overwhelmingly
negative: in a survey of over 400 citizens older than 50 years in 2016, approximately 92% considered
their socio-economic conditions to be better under pre-dam natural flows (Baah-Boateng et al., 2017).
The costs borne by the river ecosystem and the communities in the vicinity of the dams in the Lower
Volta is in strong contrast to the vast economic benefits that Ghana as a whole has enjoyed from the
relatively affordable hydropower, irrigation schemes and tourism that developed after construction
of the dams (Eshun and Amoako-Tuffour, 2016; Alhassan, 2009). After construction, the Akosombo
Dam provided over 70% of Ghana's electricity and is credited with powering Ghana's industrialization
and making it one of the most developed countries in West Africa (Alhassan, 2009). The dam currently
makes up about 20% of the installed electricity generating capacity in Ghana (Dye, 2020) and is
operated by the Volta River Authority (VRA).
The local-national mismatch in benefits deriving from the operation of the Akosombo dam has been
investigated in previous studies, notably in 2016 by Ntiamoa-Baidu et al. (2017). The 2016 study
adopted a simulation based approach using the Water Evaluation and Planning (WEAP) tool to
compare the current flow regime with the natural flow regime and two other scenarios for re-
operating the Akosombo dam (Annor et al., 2017; Mul et al., 2017). The alternative dam operation
scenarios were found to reduce power generation by 45% to 74%, which were deemed undesirable
(Annor et al., 2017). This was at a time when Akosombo and Kpong dams made up about 40% of
installed capacity and Ghana was experiencing power rationing due to low water levels in the Volta
River and a shortfall in gas supply to other power plants (Dye, 2020). The present energy context of
Ghana is very different and is characterized by an "overabundance" of electricity generation potential
- almost twice the peak load demand and therefore a reduced dependence on power generation at
Akosombo and Kpong dams. (Dye, 2020; Kumi, 2017). While installed capacity does not directly
translate into power delivery, it is worth re-examining the trade-offs between water users in the Lower
Volta under this changed situation given that it is as a result of 'take-or-pay' power purchase
agreements with private power companies whereby 90% of the power made available by these



companies has to be paid for irrespective of whether it is used or not (Dye, 2020). In 2018, the cost of
this extra capacity was approximately 5% of the country's gross domestic product (GDP) (The World
Bank, 2018; Dye, 2020).

## 3   Methods

The simulation model for this study was developed using the mass balance of inflows, net evaporation
rates and releases from the Akosombo Dam from 1981 to 2012 (data obtained from VRA). In addition
to net evaporation and inflows, additional input data to the model consisted of downstream water
levels and physical characteristics of the dam such as the storage-area-level relationships. The model
was initially set up using the current baseline dam operation policy for hydropower, flood control and
irrigation over a wet (2010), dry (2006) and normal (1985) year. The Nash-Sutcliffe model efficiencies
of 0.89, 0,91 and 0.90 respectively were obtained when the modelled and observed reservoir volumes
were compared for each of the wet, dry and normal years, thus indicating a good model fit (Figure S1-
S3, Supplementary material).
Radial Basis Functions (RBF) were used to parameterize the control policies for mapping reservoir
levels and the time into daily release decisions (Zatarain Salazar et al., 2016, 2017). RBFs are non-linear
approximating networks that allow the time dependent operating decisions to depend on more than
a single variable and most importantly can accommodate multiple objectives simultaneously (e.g.,
hydropower, irrigation, flood prevention, and e-flows). By providing alternative Pareto-optimal
solutions, it is possible to visualise trade-off between the objectives and thereby inform policy
decisions. In a comparative analysis, Giuliani et al. (2016) found that RBF solutions performed better
in terms of convergence, consistency and diversity of solutions as compared to another widely used
universal approximator, Artificial Neural Networks (ANN). Indeed, using such a non-linear
approximating network avoids "restricting the search for the optimal policy to a subspace of the
decision space that does not include the optimal solution" (Giuliani et al. 2018).

### 3.1   Multi-objective problem formulation for the Lower Volta system

When the storage volume in the reservoir is known at time $t$, and the decision time step is set to 1,
the downstream releases can be determined for the time interval $[t, t+1]$. The release from
Akosombo Dam ($r_{t+1}^{Ak}$) is determined by the irrigation, hydropower, environmental and flood control
demands ($r_{t+1}^{IHEF}$) (Eq. ( 1 )) (Figure 2). Kpong Dam is operated as a run-of-the-river hydropower system
by VRA after water is diverted for irrigation (Eq. ( 2 )).





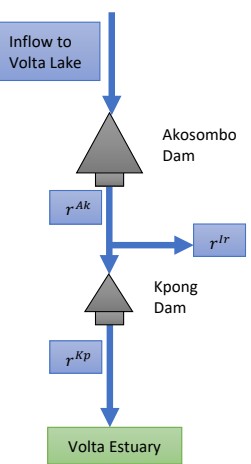


*Figure 2: Topology of reservoir system in the Lower Volta. $r^{Ak}$, $r^{Kp}$ and $r^{Ir}$ are the flow releases from Akosombo, Kpong and for irrigation respectively.*

The release from Kpong Dam ($r_{t+1}^{Kp}$) is therefore calculated as the difference between the release from Akosombo Dam and irrigation ($r_{t+1}^{Ir}$), and represents the downstream releases for hydropower, e-flows and floods ($r_{t+1}^{HEF}$).

$$r_{i+1}^{Ak} = r_{i+1}^{IHEF} \qquad (1)$$

$$r_{i+1}^{Kp} = r_{i+1}^{Ak} - r_{i+1}^{Ir} = (r_{i+1}^{HEF}) \qquad (2)$$

The operating policy is commonly parametrised as a function of the reservoir storage volume at a particular time. The parameterized operating policy $f$ is then defined as a mapping between the decisions $\boldsymbol{u}_t$ and the policy inputs $z_t$ comprising the time $t$ and system state, or storage volume, $x_t$, Eq. ( 3 ), namely:

$$\boldsymbol{u}_t = f(z_t) \qquad (3)$$

The $k$th decision variable in the vector $\boldsymbol{u}_t$ (with $k = 1, \dots, n$) is therefore defined as a weighted sum of radial basis functions, as specified in Eq. ( 4 ):

$$u_t^k = \sum_{i=1}^{N} w_{i,k} \varphi_i(z_t) \qquad (4)$$

where N is the number of radial basis functions $\varphi_i(\cdot)$, and $w_{i,k}$ is the non-negative weight of the $i$th radial basis function, and the N weights sum to unity. The $i$th Gaussian radial basis function is then given by Eq. ( 5 ):


$$\varphi_i(z_t) = exp\left[ -\sum_{j=1}^{M} \left( \frac{(z_t - c_{j,i})^2}{b_{j,i}^2} \right) \right] \qquad (5)$$

Where $j = 1, .., M$ is the number of input variables, $z_t$ is the policy input (e.g. time $t$, reservoir level
$x_t$) and $c_{j,i}$ and $b_{j,i}$ are the centres and radii respectively of the $i$th Gaussian radial basis function for
the $j$th input variable. The parameter vector $\theta$ is defined as $\theta = (c_{i,j}, b_{i,j}, w_{i,k})$ with $i = 1, .., N$; $j =$
$1, .., M$; $k = 1, .., n$, where the centre and radius are normalized with $c_{i,j} \in [-1,1]$ and $b_{i,j} \in (0,1]$.
The policy parameters $\theta$ are determined by simulating the system over the time horizon $H$ under the
policy $f = \{f(t, x_t, \theta): t = 0, \dots, H - 1\}$. In this way the inputs to the RBF policy (time index and
reservoir storage volume) are mapped to the outputs (release decisions). The policy parameters, are
evaluated by solving the multi-objective problem function, $f$, specified in Eq. ( 6 ) in the objective space
using an informed search algorithm $\theta^*$. The objective functions $J$ are the operating objectives of the
reservoir as defined in Eq.s ( 7 ) to ( 12 ) with any maximization objectives multiplied by -1 to
reformulate all the objectives as minimizations.

$$f\theta^* = \arg \min_{\theta} J(\theta) \qquad (6)$$

The number of RBFs used in this study was four (i.e. $n = 4$). Thus, the total number of parameters ($\theta$)
for the control policy in this study is 24. A daily decision timestep, representative of "real operations"
of the two dams by VRA was used (Annor et al., 2017). The simulation time horizon, $H$, of 29 years
using historical data starting from January 1984, was constrained by the availability of data. This period
however encompasses key dry and wet periods. In 1997-2000 and 2006-2007, there was power
rationing in Ghana as water levels in the Akosombo reservoir fell to 73.01 m (July 1999) and 72.16 m
(August 2006) respectively; both lower than the minimum operating level of 73.15 m (VRA, 2021). On
the other hand, in 1991 and 2010, extremely high inflows caused the reservoir water level to rise to
83.90 m and 84.42 m, respectively, close to the maximum operating level of 84.73 m, and necessitated
the opening of the spillways. The 2010 reservoir level remains the highest point ever recorded at
Akosombo. The four objectives considered in this study are described in more detail below:
1.  Annual hydropower: Maximization of the annual hydropower generated at Akosombo and Kpong
dams as defined in Eq. ( 7 ). While the annual firm power requirement from Akosombo Dam is 4415
GWh/year, the amount of electricity generated has typically exceeded this target in the past due
to high national dependence on power generation from this dam. As such, operations at Akosombo
has generally been to maximise power considering the reservoir volume and inflows to the dam
(Annor et al., 2017). There is no firm power target at Kpong which is a run-of the river dam and
generates power with releases from Akosombo after the diversion for irrigation.



$$J_H = \sum_{t=1}^{I} HP_t \qquad\qquad (7)$$

where energy production $(HP_t)$ in GW is given by Eq. ( 8 ):

$$HP_t = \eta g \rho_w h_t \, q_t^{turb} . 10^{-9} \qquad\qquad (8)$$

where $t \in I$ are the days in a year, $\eta$ is the turbine efficiency (dimensionless), $g$ is acceleration due
to gravity (9.81 m/s$^2$), $\rho_w$ is water density (1000 kg/m$^3$), $h_t$ is net hydraulic head (m) and $q_t^{turb}$ is
flow through the turbines (m$^3$/s). The hydropower objective at Akosombo is subject to constraints
on the minimum daily firm power requirement of 6 GWh/day for system stability and the maximum
possible power production due to turbine capacity (1,603 m$^3$/s) at the maximum safe operating
level of 84.12 m.
2.   Irrigation: Maximization of the volumetric reliability of water supply to meet irrigation demand as

described in Eq. ( 9 ).

$$J_I = \frac{1}{H} \sum_{t=1}^{H} \frac{q_t}{V_t} \qquad\qquad (9)$$

subject to the constraint in Eq. ( 10 ):

$$0 \leq q_t \leq Q_t \qquad\qquad (10)$$

where $V_t$ , $q_t$ and $Q_t$ are the irrigation demand, the diverted water and the flow at diversion point
at time $t$, a day within the simulation horizon $H$. The current irrigation demand is 10 m$^3$/s but there
have been plans since 2009 for this to be increased to 38 m$^3$/s for the Accra Plain Irrigation Project
and the expansion of the Kpong left bank irrigation project (GIDA, 2009). These projects are yet to
be fully realized however, and in this study the anticipated irrigation demand of 38 m$^3$/s is used as
the baseline value.
3.   Flood control: Minimization of flood occurrences defined by the average number of days where

downstream flow releases from Kpong, $r_{t+1}^{Kp}$, exceed 2300 m$^3$/s, the bank full capacity of the river

$(Q_F)$ (Eq. ( 11 )). Opening of the spillways of the Akosombo and Kpong dams is quite rare and has

occurred only twice, in 1991 and 2010. Consequently the riparian communities are ill-prepared for

flood releases and incur high losses whenever floods are released (Ayivor and Ofori, 2017).

$$J_F = \frac{1}{H} \sum_{i}^{H} \left( \frac{\max(r_{i+1}^{Kp} - Q_F, \, 0)}{Q_F} \right) \qquad\qquad (11)$$

4.   E-flows: The trade-off between the three objectives defined above and three different e-flows

(Figure 3) are investigated in separate runs in this study. As such, three different configurations of

the trade-off problem are investigated.



a.  Clam e-flows (Figure 3a): This e-flow was designed for the Lower Volta River using the Volta
clam, a stenotopic, freshwater bivalve, as an indicator species (Owusu, Mul, Strauch et al.,
2022). The recommended flow is a low flow range of 50 - 330 $m^3/s$ from November to March
to support the clam's veliger larvae stage, a key life stage in its lifecycle. An 80% reliability of
this flow occurring in the stipulated months is an acceptable compromise for the survival of
the clam veliger larvae (Owusu, Mul, Strauch et al., 2022). While only a low flow is prescribed
for five months in this e-flow recommendation, this necessarily implies that flow releases at
other times of the year will be higher, although the magnitude, duration and timing of this such
flow is not defined and thus does not form a constraint for clam e-flows. The historical
minimum for high flows in September and October under pre-dam flows was 1052 $m^3/s$.

b.  Natural flow dynamics considering bank full flows (e-flows 2) (Figure 3b): This e-flow reinstates
natural flow dynamics in the Lower Volta to support multiple ecosystem services including
fisheries, aquatic weed control, flood recession agriculture and sediment transport (Mul et al.,
2017). The minimum discharge for the high flow period in September to October is 2,300 $m^3/s$,
(which is the bank full flow rate) to ensure that river overtopping and thus some minimum
flooding of pre-dam floodplains occurs. The maximum dry season discharge for the rest of the
270          year is 700 $m^3/s$.

c.  Natural flow dynamics considering future dry season irrigation (e-flow 3) (Figure 3c): This e-
flow also re-instates natural flow dynamics of the Lower Volta while providing water for future
dry season irrigation demands (Mul et al., 2017). The minimum high flow in September and
October is 3000 $m^3/s$, which inundates an area of approximately 156 $km^2$, to support creek
fishing and flood recession agriculture. The maximum dry season flow rate is 500 $m^3/s$.

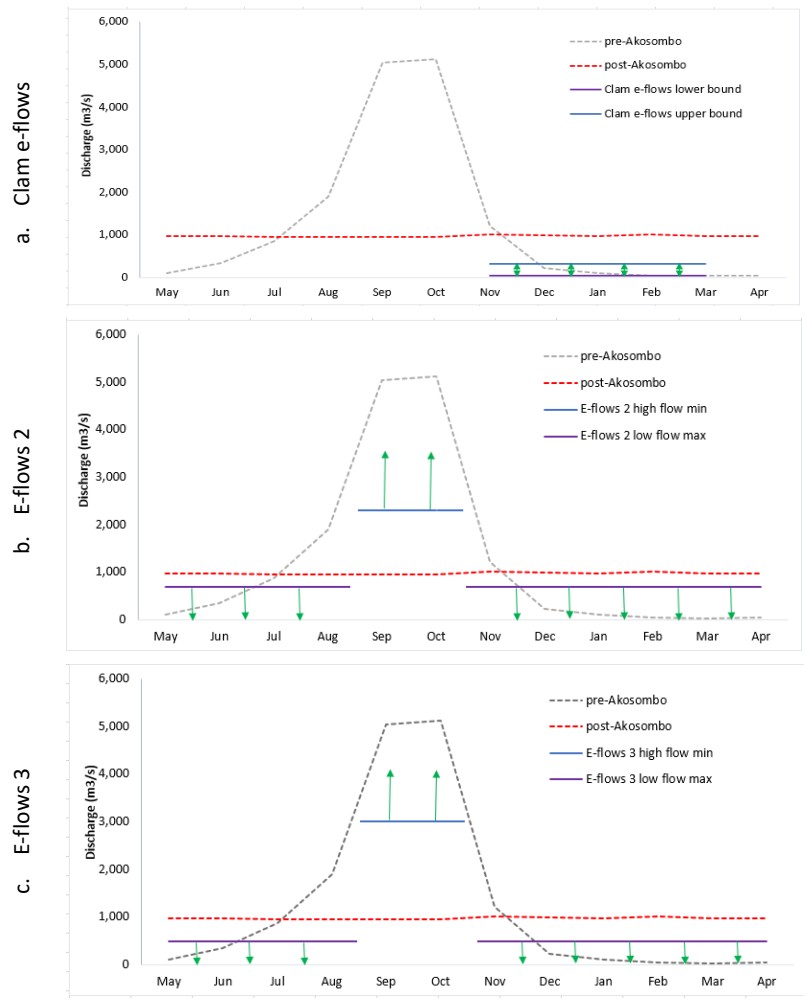

*Figure 3: E-flow configurations considered in this study with pre-dam (natural) and current post dam flow regime in the Lower Volta provided for comparison (using monthly average flow data from Volta River Authority, Ghana) for a hydrological year which starts in May in the Lower Volta. For clam e-flows the green arrows show the range of low flows recommended from November to March. For e-flows 2 and 3, the prescribed flow for September and October are a minimum threshold while for the other ten months, this flow is a maximum threshold. The green arrows begin at these thresholds and point in the direction of where the flows should be per the e-flow recommendation.*

The three alternative e-flow objectives (clam e-flows, e-flows 2 and e-flows 3) were modelled as a maximization of the reliability of the recommended flow rates occurring (Eq. ( 12 )):

$$J_E = 1 - \frac{n_E}{n_T} \qquad (12)$$

where $n_E$ is number of days when downstream flow falls outside the e-flow range and $n_T$ is the total number of days when the recommended e-flows are required.



## 3.2 Future climate scenarios

In addition to a baseline scenario optimizing the trade-offs between hydropower, irrigation, flooding and e-flows under the present climate, future scenarios representing different climate futures were analysed. The recent Sixth Assessment Report of Working Group I of the Intergovernmental Panel on Climate Change (IPCC) projects that mean temperature in West Africa will increase 1.5 °C by 2040 and projects with high confidence that monsoon rainfall over West Africa will increase in the mid (2041-2060) to long term (2081-2100) but have a delayed start (IPCC, 2021). Future projections made with medium confidence relate to the delayed retreat of the monsoon rains and an increase in the frequency and duration of droughts in the latter part of the 21st century (IPCC, 2021).

These latest climate projections draw a mixed picture of future climate in the West African sub-region. A further review of the anticipated impacts of climate change specifically on runoff in the Lower Volta was carried out with the goal of identifying studies that focussed on the entire Volta basin, either as a whole or all sub-basins. A search was conducted in Scopus using the search string: TITLE-ABS-KEY (Volta AND climate AND (change OR impact), AND (flow OR discharge OR water) AND (availability OR resources)). From the 60 papers returned, a review by Roudier et al. (2014) on climate change impacts on runoff in West African Rivers provided the first point of reference. From this review by Roudier et al. (2014), four studies meeting the search criteria were identified. An additional four papers from the Scopus search results, not reviewed by Roudier et al. (2014), were also retained for analysis.

Four papers (Kunstmann and Jung, 2005; Aerts et al., 2006; Jung et al., 2012; Abubakari, 2021) found that annual runoff in the Volta will increase (by 4% to 65%), and of these papers, three (with the exclusion of Aerts et al. (2006)) also presented monthly trends which generally showed an increase in wet season flow from June to October and a decrease in dry season flow. The findings on only monthly trends of Jin et al. (2018) are also in line with these predictions. McCartney et al. (2012) and Sood et al. (2013), in contrast, find that there will be a decrease in annual runoff (ranging from -13% to -45%) while Amisigo et al. (2015) find that the results across the various scenarios are inconsistent. Table S1 in the supplementary materials provides further details on the papers, the climate change scenarios considered and the models used. It is important to note that using a combination of models, i.e.: global and regional climatic models and then hydrological models, introduces uncertainty in the findings of climate projections for runoff (McCartney et al., 2012) and the wide ranging results from the reviewed papers show that particularly in the case of the Volta the direction of change in runoff is still unclear.

The climate-runoff studies, just as the latest IPCC report, present a mixed picture for the Lower Volta. Therefore, bearing the high level of uncertainty in mind, five scenarios indicative of the range of climate-induced changes predicted for the Volta discharge for the mid to long term are investigated

(Table 1). These include both increases and decreases in annual runoff as well as seasonal variations
in runoff into the Lake Volta.
*Table 1: Design of future scenarios encompassing climate-induced changes in the Lower Volta discharge*

| Scenario | Annual decrease | Annual increase | | Seasonality | | |
|---|---|---|---|---|---|---|
| | Decrease -45% | Increase +12% | Increase +65% | Dry season decrease (Nov to May) -10% | Wet season increase (Jun to Oct) +10% | Wet season increase (Jun to Oct) +55% |
| 1 | x | - | - | - | - | - |
| 2 | - | x | - | - | - | - |
| 3 | - | - | x | - | - | - |
| 4 | - | - | - | x | x | - |
| 5 | - | - | - | x | - | x |

## 323    4   Results

The relationship between different water users in the Lower Volta is presented using parallel axis plots
(Figure 4 and Figure 5). Every line crossing the axes is a Pareto-approximate (non-dominated) solution
and shows the performance of each water user under an alternative dam operation policy for the
Lower Volta system. The range of values for each water user has been normalized using its maximum
and minimum values with the best performance featuring at the top of each axis. For irrigation and
the environmental objectives, the highest value is interpreted as a dam operation policy whereby
100% of irrigation demand and e-flows are provided, while for the flood control objective, this is
interpreted as there being no downstream flow releases above the flooding flow threshold of 2300
$m^3$/s over the simulation horizon. For hydropower at Akosombo and Kpong dams, the maximum and
minimum values used in the normalization encompass the maximum and minimum annual
hydropower generated across all the scenarios considered: at Akosombo Dam these are 5,100 and
845 GWh/year respectively and at Kpong Dam, 1,000 and 130 GWh/year respectively. The trade-offs
between the three e-flow objectives and other water users are shown with the 'best' or highest
performing operation policy for each objective highlighted. Additionally, room for compromise,
characterised in this study as operation policies meeting the firm hydropower demand of 4,415
GWh/year for Akosombo Dam ("fair hydropower") and e-flow demands at least 80% of the time ('fair
environment'), have also been highlighted. It is important to note that, the terms 'best' and 'fair' as
used here, are not qualifying adjectives but solution descriptors in the Pareto approximate space with
the latter used to denote 'reasonable' or 'satisfactory' rather than 'equitable' solutions. Cumulative
distribution graphs showing the function values for the baseline and future scenarios are presented in
Figure S4 in the supplementary materials.





### 4.1 Baseline scenario


For the baseline scenario (Figure 4), the highest performing dam operation policies for hydropower
trade-off sharply with the provision of e-flows (all configurations) in the Lower Volta such that there
is no overlap even among fair solutions for either objective for all e-flow configurations considered in
this study. To meet the current firm energy requirement of 4,415 GWh/year at Akosombo, e-flows can
only be released about 60%, 49% and 47% of the required time for the clam e-flows, e-flows 2 and e-
flows 3, respectively. From the environmental perspective, the results show that hydropower demand
from Akosombo and Kpong have to fall to approximately 3,903 GWh/year and 760 GWh/year
respectively for the release of clam e-flows; or approximately 3000 GWh/year and 563 GWh/year for
the release of e-flows 2; or alternatively approximately 2711 GWh/year and 508 GWh/year for the
release of e-flows 3 to become possible 80% of the recommended time under current climatic
conditions. The solutions for clam e-flows generally lead to higher hydropower generation compared
to the other e-flows because for seven months of the year there is no constraint on water releases in
the Lower Volta for the environment, hence hydropower generation can be maximised in these
months. Comparing the dynamic e-flow configurations, e-flows 2 yields higher hydropower generation
because its dry season flow recommendation is higher at 700 $m^3$/s as compared to that of e-flow 3
which is 500 $m^3$/s thus allowing for higher power generation in the dry season.
Considering the relatively low water demand for irrigation as compared to hydropower, marginal
increases in hydropower generation at Kpong lead to significant reduction in the amount of irrigation
demand that is met in the baseline scenario. The solutions for all e-flow configurations perform well
for the flood control objective even though e-flow 2 and 3 prescribe flood releases for two months of
the year. As such, comparing clam e-flows to e-flows 2 and 3, there is a reduction (0.99 for clam e-
flows vs. 0.83 for e-flows 2 and 3) in the performance of the 'best' solution for the latter two, as
expected, showing that the requirement for floods for two months in a year in those e-flow
configurations are not being met.

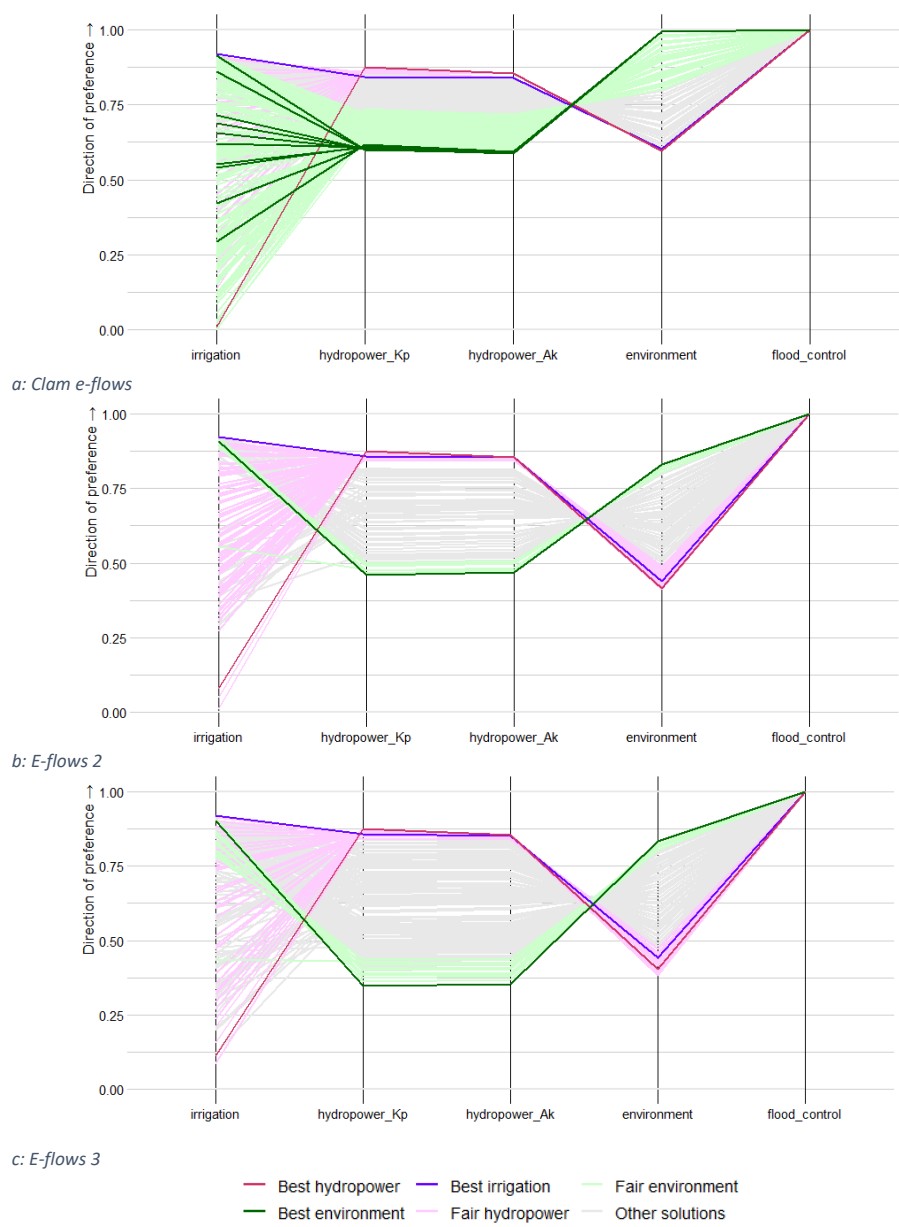

*Figure 4: Full set of non-dominated solutions in the baseline scenario with the 'best' solution for each water user and 'fair' solutions for hydropower and environment highlighted. The objective values for each water user have been normalised using the minimum and maximum values over all simulations with the highest performance for each objective placed at the top of the axes. The order in which water users are presented has been chosen to highlight trade-offs between them.*

## 4.2 Future scenarios


The effects of different climate futures on the Pareto-optimal solutions for the Volta basin are
presented in Figure 5. Scenario 1, where there is 45% decrease in annual inflows to Akosombo dam


stands out as the system becomes water stressed so that the best performing operation policies even
for hydropower lead to only about 2776 GWh and 550 GWh annual power generation at Akosombo
and Kpong respectively for all e-flow configurations. The best operating policies for the environment
however improve slightly from 0.99 to 1 for clam e-flows and remain unchanged at 0.83 for e-flows 2
and 3 relative to the baseline. This is because these solutions, even in the baseline scenario are those
for which only dry season low flows are released. In contrast to Scenario 1, under Scenario 3 and to a
lesser extent Scenario 2, where annual flows to the Akosombo Dam increase by 65% and 12%
respectively, the solutions move up on the two hydropower axes and some fair solutions for the
environment lead to higher annual hydropower production of up to 4,242; 3,392 and 2,926 GWh/year
for clam e-flows, e-flows 2 and e-flows 3 respectively at Akosombo.
Seasonal climate change effects on the Lower Volta system under scenarios 4 and 5 are comparable
to annual climate change effects. As a result, the solutions for Scenario 4 are similar to Scenario 2
while those for Scenario 5 are similar to Scenario 3 save for the slightly lower hydropower generation
values for Scenario 5 and hence fewer 'fair hydropower' solutions in line with its relatively lower
inflows (+65% annual inflows for Scenario 3 vs 55% wet season inflows for Scenario 5). This is due to
the high residence time of water in the Lake Volta (3.9 years) and the fact that the Lower Volta has
highly seasonal inflows naturally so that an annual inflow increase applied to inflows across the year,
as applied in Scenario 2 and 3, results in a minimal increase in the absolute values of inflows to the
dam in the dry season but a relatively significant increase in the absolute values of wet season inflows,
thus amplifying seasonality.





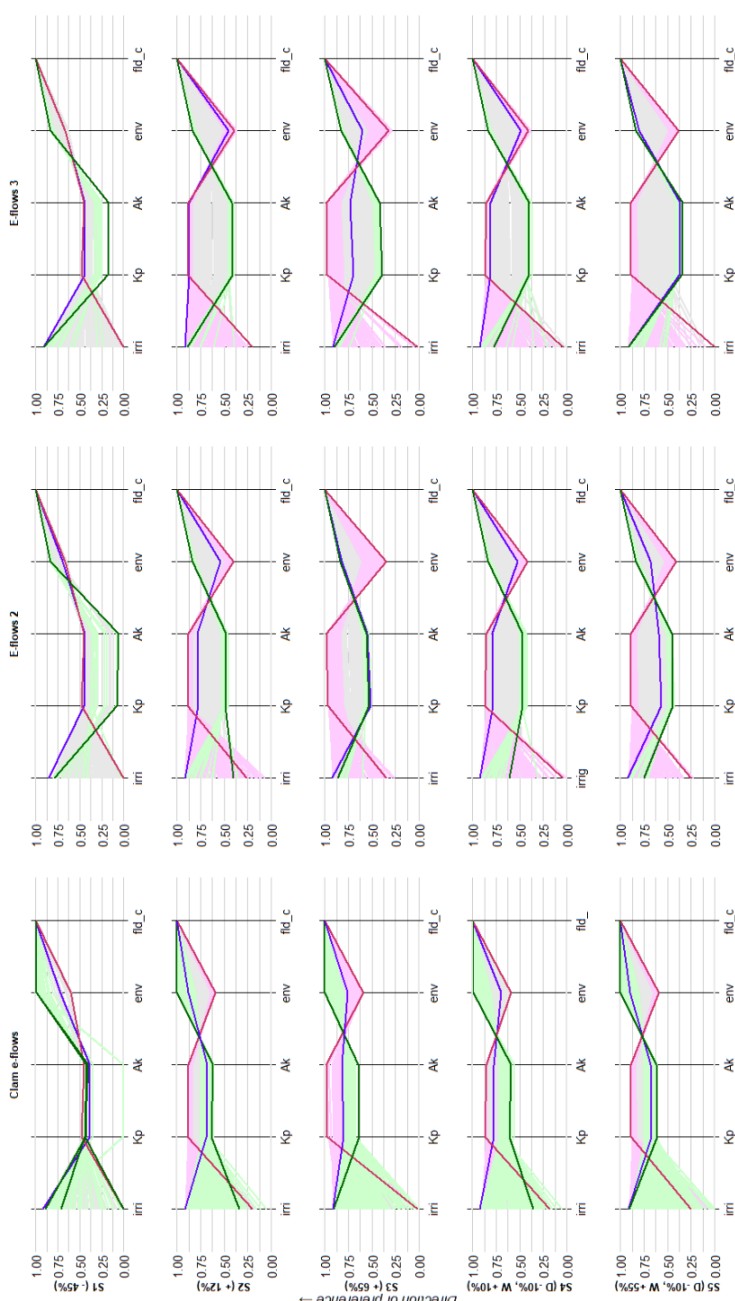

*Figure 5:* Full set of non-dominated solutions in the future scenarios with the 'best' solution for each water user and 'fair' solutions for hydropower and environment highlighted. The objective values for each water user have been normalised using the minimum and maximum values over all simulations with the highest performance for each objective placed at the top of the axes. The order in which water users are presented has been chosen to accentuate the trade-offs. Notation- water users: Irri- irrigation, Kp- hydropower from Kpong Dam, Ak-hydropower from Akosombo Dam, env- environment (clam e-flows, e-flows 2 or e-flows 3), fld_c- flood control. Notation- scenarios: S1 to S5- Scenario 1 to 5, D- dry season flow, W-wet season flow.


## 5   Discussion


The Lower Volta River System is characterized by the dominance of hydropower generation for Ghana
and its neighbouring countries. This has come at a high cost to downstream ecosystem services and
communities (Lawson, 1972; Tsikata, 2008; Ntiamoa-Baidu et al., 2017). The results from this study
show that likewise, some cost to hydropower production would have to be borne for e-flows
implementation and the restoration of some of these ecosystem services under current climatic
conditions. For the implementation of clam e-flows 80% of the time, i.e. a fair environmental solution,
the country would forfeit at least 11.6% of the firm annual power demand from Akosombo Dam. For
the implementation of only the dry season flow recommendations of e-flows 2 and 3, about 32% and
38% respectively of current firm energy requirement would have to be supplemented with power
generation from other sources. The release of floods as recommended in the dynamic e-flow
configurations (e-flows 2 and 3) is not a Pareto-optimal operation policy within the current operating
constraints of the Akosombo Dam because in addition to flooding  pre-dam floodplains which are now
permanently inhabited (Ayivor and Ofori, 2017), releasing flows above the maximum turbine capacity
of 1,603 m$^3$/s at Akosombo means that these water volumes are lost to power generation. Flood
releases also far exceed irrigation water demands.
While the majority of climate predictions for the Volta River generally point to an increase in annual
water availability (Kunstmann and Jung, 2005; Aerts et al., 2006; Jung et al., 2012; Abubakari, 2021;
Jin et al., 2018; Sylla et al., 2018), based on this study, an argument can be made that both an increase
or a decrease in inflows to the Lower Volta enhance the potential for e-flows implementation
compared to the current baseline. On the one hand, an increase in inflows to the Akosombo dam as
applied in scenario 3, reduces the amount of the firm energy requirement that would have to be
supplemented by other sources for the implementation of 'fair environmental solutions' to about
3.9% (vs 11.6%) for clam e-flows, and then 23.2% (vs 32%) for e-flows 2 and 33.7% (vs 38%) for e-flows
3. On the other hand, a decrease in inflows to the Akosombo Dam, whereby at best only 2,774
GWh/year of hydropower can be generated, provides opportunity to strategically release
recommended dry season e-flows to reap some environmental benefits out of a 'bad' situation where
annual flow releases from the dam will be low anyway. This operation policy under dry climate
scenarios could also be adopted in dry years, in essence modelling the Episodic E-flows
Implementation approach, which is an opportunistic approach to dam re-operation that takes
advantage of prevailing hydrological conditions (Warner et al., 2014; Yang & Yang, 2014; Owusu et al.,
2021). This contrasts with the alternative approaches, Adaptive Management and Blanket Operation
which represent more structural inclusion of e-flows in the dam operation policy (Warner et al., 2014).



Only future climate scenarios were modelled in this study; however, inferences can also be made on
the effect of simple energy and water demand futures on the Lower Volta system. For instance, an
increase in irrigation demand will trade-off against hydropower production at Kpong Dam and an
increase in the firm energy requirement or the continuation of the *de-facto* policy of hydropower
maximisation at Akosombo Dam, despite the availability of alternative power generation sources
(Dye, 2020; Kumi, 2017), will weaken the potential for re-operation of the dam for the riverine
environment. Changes in upstream water consumption as well as the construction of new dams such
as the Pwalugu Dam in northern Ghana will also affect inflows to the Akosombo Dam. Gonzalez et al.
(2021), however, show that practical coordination of the operation of major infrastructure in the Volta
Basin, as compared to the current approach whereby dam operators fail to consider downstream built
infrastructure, reduces the impact on inflows to the Akosombo Dam in particular, and also maximises
basin-wide benefits. Undoubtedly this coordination should extend beyond the Volta Basin to include
the entire electricity generation portfolio of Ghana to further reduce the impact of e-flows
implementation in the Lower Volta on power supply in the country.
In the potential re-operation of the Akosombo and Kpong dams, one has to consider that the majority
of the alternative sources of power in Ghana use carbon fuels (Dye, 2020) and thus most likely
contribute more to climate change compared to power generation from these two existing dams (dos
Santos et al., 2006; Barros et al., 2011). However, the potential re-operation of the Akosombo and
Kpong dams can benefit from (i) the groundwork laid by research on the pre- and post-dam river
system (Lawson, 1972; De-Graft Johnson, 1999; Tsikata, 2008; Adjei-Boateng et al., 2012; Obirikorang
et al., 2013; Nyekodzi et al., 2018; Owusu et al., 2022), (ii) insights deriving from interviews and
extensive stakeholder engagement (Ayivor and Ofori, 2017; Ohemeng et al., 2017; Nukpezah et al.,
2017), and (iii) existing supporting legislation for e-flows implementation (Anon, 2001). Indeed,
research on successful and stalled cases of dam re-operation indicates that stakeholder engagement
and supporting legislation enhance the chances of successful e-flows implementation (Owusu et al.,
2021; Owusu, Mul, van der Zaag et al., 2022) .
## 6   Conclusion
A dam is designed with future uses in mind – this provides the justification for its construction. The
future, however, can turn out differently from that envisaged in the dam design. Therefore, re-
operation of the dam to meet changing demands is a likely necessity. This study investigates current
and future trade-offs between water users in the Lower Volta River Basin and specifically explores the
potential to deliver environmental flows to support various ecosystem services that have been
negatively impacted by the current operation of the Akosombo and Kpong dams. The results highlight



the dominance of hydropower production in the Lower Volta; if this relaxes there is more opportunity
for restoration of the riverine environment under current climate and water use conditions and even
more so under future scenarios where inflows to the Lake Volta increase. In future scenarios whereby
inflows to the Lake Volta decrease, it is still possible to strategically manage and time water releases
to provide dry season low flows which will support the clam fishery and help control aquatic weeds
and some water borne diseases in the Lower Volta.  This study applied advanced optimisation
techniques to identify and analyse dam operation policies for e-flows under discreet climatic
scenarios. Future studies should focus on the robustness and limits of these policies under
multitudinous future climatic and water use scenarios. Such robustness studies, together with flow
experimentation, will reveal dam operation policies that may be adopted with some confidence
presently. It will also build on groundwork already laid, through e-flows legislation and extensive
collaborative scientific studies, for the successful re-operation of the Akosombo and Kpong dams for
the environment and other water users.

**Acknowledgement**

This research is part of the EuroFLOW project (EUROpean training and research network for
environmental FLOW management in river basins) funded by the European Union's Horizon 2020
Research and Innovation Programme under the Marie Skłodowska-Curie grant Agreement (MSCA) No.

478    765553.

J.Z.S and J.S. are supported by the Multi-Actor Systems Research Programme of TU Delft.

**Data availability**

The data associated with this manuscript will be made available upon consultation with the national
organisation, Volta River Authority (VRA), who owns the data

**Declaration of interests**

Some authors are members of the editorial board of Hydrology and Earth Systems Sciences journal.
The authors declare that they have no other known competing financial interests or personal
relationships that could have influenced the work reported in this paper.

**Author contribution**

AO: Conceptualization, Methodology, Data collection, Formal analysis, Interpretation, Writing-
original draft preparation;
JZS: Methodology, Formal analysis, Writing-review and editing;





491 MM: Conceptualization, Data collection, Writing-review and editing, Supervision, Project

492 administration;

493 PvdZ: Conceptualization, Writing-review and editing, Supervision, Funding acquisition;

494 JS: Conceptualization, Methodology, Writing-review and editing, Supervision, Funding acquisition





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
