# Peer review of "Quantifying the trade-offs in re-operating dams for the environment"

_Hydrology and Earth System Sciences, 2022_

## Referee Comment (RC1)

Title: Quantifying the trade-offs in re-operating dams for the environment in the Lower Volta River.

**Overall summary**

The manuscript quantifies the trade-offs between environmental water uses and hydropower generation in the Lower Volta River Basin using Multi-objective Direct Policy Search. The topic is of general interest given the worldwide boom expansion of new hydropower dams and the environmental and social impacts of dam construction and operations. However, the manuscript's contribution is not clearly presented, nor are the broader implications for Ghana of hydropower reduction discussed.

In terms of the manuscript's contribution is not clear if the authors are presenting a methodological advance or presenting the system trade-offs among different water users in the Lower Volta River basin. If the contribution is methodological, it is not clear how what the author present is different from other previous studies that use EMODPS. If the contribution it is to presents the system trade-offs, the discussion/implications of reducing hydropower generation lack broader consequences for the country and its economy. How could the results be presented to relevant decision-makers in the country if only local 'benefits' (lines 444 to 452) are given to 'validate' the implementation of hydropower reduction to support e-flow implementations? Does Ghana have alternatives to replace cheap and flexible hydropower generation? If the reduction in hydropower is replaced by thermal generation, how will this impact nationally determined contributions (in terms of $CO_2$ emission reduction)? What is the local vs country-wide trade-offs of reducing hydropower generation? I consider these questions need to be answered if the paper aims to be a viable input to any discussion on re-operating the Akosombo and Kpong hydropower plants. I recommend improving the paper's discussion and implications by including a clear discussion of the previously highlighted points.

It is not clear how robust are the presented results in terms of future climate change impacts or upstream water uses changes. The Akosombo and Kpong hydropower plants are located downstream of the Volta River basin, and any change in water availability upstream in the basin could possibly impact the system evaluated in this study. Methods such as Sensitivity Analysis or Exploratory modelling evaluating 'a wider range' of future states of the system could provide more conclusive results and improve the discussion of future impacts of changes in water availability.

I consider the manuscript should be reconsidered after major revisions.

**Detailed comments**

1) Lines 23 to 24, "There is uncertainty in climate change effects on runoff in this region." The study evaluates the impact of six climate change scenarios. Are those six scenarios representative of and account for the climate change uncertainty in the Volta River basin? How robust are the presented results of the six evaluated scenarios?

2) The literature review only focuses on EMODPS; there is no discussion about other methods used to identify reservoir operations. How does EMODPS compare with other techniques? Also, there is no transition between presenting MOEAs and EMODPS methods. MOEAs should be presented first, as EMODPS uses MOEAs to identify reservoir policies.

3) Lines 92 to 93. "As such, the implications of the trade-off on power delivery, energy prices and carbon emissions are not investigated". Dams and hydropower plants are not isolated infrastructures. The state-of-the-art is moving to evaluate the multi-sector implications of human-nature resources systems. Any reduction in hydropower generation not only impacts "the water demands" for this use, its impacts the power system (emission, energy prices, country economy, etc.). I recommend improving the discussion on the real implications of reducing hydropower generation in the country.

4) Lines 157 to 165. Is it unclear why the model was calibrated for three years with specific conditions to the system (wet, dry and normal) if the authors have access to historical data from 1981 to 2012? Why do no to calibrate the model for the full historical time-series? The calibration process produced only one reservoir operating policy for the wet, dry and normal conditions of the system, or it produces a policy by condition.

5) Line 116 "to a steady flow of about 1,000 m3/s all year round". Is this steady flow the average flow downstream of Akosombo? Could the authors inform about the Akosombo release seasonality and monthly flow variability?

6) Lines 224 to 225. "While the annual firm power requirement from Akosombo Dam is 4415 GWh/year." [GWh] units refer to energy while [GW] units to power. Please, check if you are referring to "firm energy generation".

---

## Author Comment (AC2)

**Quantifying the trade-offs in re-operating dams for the environment in the Lower Volta River**

Afua Owusu[a b], Jazmin Zatarain Salazar[b], Marloes Mul[a], Pieter van der Zaag[a c], Jill Slinger[b d]

[a] Land and Water Management Department, IHE Delft Institute for Water Education, Westvest 7, 2611 AX Delft, The Netherlands
[b] Faculty of Technology, Policy and Management, TU Delft, Jaffalaan 5, 2628 BX Delft, The Netherlands
[c] Faculty of Civil Engineering and Geosciences, TU Delft, Stevinweg 1, 2628 CN Delft, The Netherlands
[d] Institute of Water Research, Rhodes University, Drosty Rd, Grahamstown, 6139, South Africa

**Response to reviewers:**
The authors thank the reviewers for their insightful comments which have proved very helpful in revising the manuscript. Below are detailed responses to the individual comments raised by each reviewer with new/modified text highlighted. In responding to reviewer 1, the specific comments are addressed before the response to the questions raised in the overall summary.

| Location | Comment and Response |
|---|---|
| **Reviewer 1** | |
| Lines 23 - 24 | **Comment 1.1:**

"There is uncertainty in climate change effects on runoff in this region." The study evaluates the impact of six climate change scenarios. Are those six scenarios representative of and account for the climate change uncertainty in the Volta River basin. How robust are the presented results of the six evaluated scenarios?

**Response:**

In section 3.2, the systematic literature review which informed the choice of the *five* runoff scenarios is presented. This literature review identified papers that specifically focussed on the anticipated impacts of climate change on run-off in the entire Volta Basin, either as a whole or all sub-basins. These climate-runoff studies presented a very varied and thus uncertain picture with predicted changes in annual runoff ranging from +65% to -45%. The five scenarios presented in this paper span across this range for annual and seasonal runoff.
The high uncertainty in climate projections for the study area is a major challenge in developing adaptation strategies for the region. However, in this study, the conclusion from the analysis indicates that both an increase or decrease in inflows (due to climate change) lowers the trade-off between e-flows and hydropower. The authors believe that this is an important point to make as it shows that with respect to the existing e-flows recommendations, uncertainty about the direction of CC is not a limitation to their potential implementation.
On the robustness of the presented results for the 5 evaluated scenarios, in the conclusion the authors recommend that "*Future studies should focus on the robustness and limits of these policies under multitudinous future climatic and water use scenarios*.". We want to emphasize that our goal for this study was to first discover Pareto optimal policies across the objectives |

| Location | Comment and Response |
|---|---|
| **Reviewer 1** | |
| | of the Volta River basin under the five plausible scenarios. Nonetheless, as a follow up to this present study, an uncertainty evaluation has now been carried out for the optimised release policies found in this study for 2400 scenarios and the robustness for each policy compared across different robustness metrics (Buskop, 2022). The main results are in line with the present study in that: |
| | "Using various robustness metrics to define stakeholder preferences across objectives shows the same trade-offs as in the unmodified objective scores. Environmental policies do not perform well with energy objectives, while the hydropower policies do not work well for the irrigation and environmental objectives" (Buskop, 2022, Page vi). The findings from this MSc thesis are being prepared for journal publication.

Buskop, T. (2022). *Will the Benefits Keep Flowing?* MSc thesis, TU Delft. Available at: https://repository.tudelft.nl/islandora/object/uuid%3A02d941cf-b7a8-4498-9a13-762c7e6988fc?collection=education |
| | **Comment 1.2:**
The literature review only focuses on EMODPS; there is no discussion about other methods used to identify reservoir operations. How does EMODPS compare with other techniques? Also, there is no transition between presenting MOEAs and EMODPS methods. MOEAs should be presented first, as EMODPS uses MOEAs to identify reservoir policies.

**Response:**
The authors are grateful for this comment and have updated the introduction to first present MOEAs followed by EMODPS. The advantage of EMODPS is also presented. The changes made to the text in response to this comment in the introduction are:

Introduction:
*Multi-objective evolutionary algorithms (MOEAs) are one such tool for assessing the trade-offs between water users in a river basin. MOEAs use stochastic search tools to simultaneously find the Pareto approximate set across multiple objectives (Reed et al., 2013; Matrosov et al., 2015; Hurford et al., 2020; Zatarain Salazar et al., 2016; Kiptala et al., 2018). The Pareto approximate or non-dominated set of solutions are the suite of solutions for which increasing the water allocation to one user leads to a reduction in the benefit to others. The advantage of MOEAs is that they do not require pre-specifying preferences across objectives, thereby supporting unbiased a posteriori decision making (Reed et al., 2013; Hurford et al., 2014). Furthermore, MOEAs allow for heterogeneous and non-linear problem formulations with incommensurable objectives and different risk attitudes across objectives. Accordingly, non-market objectives can be evaluated alongside conventional economic objectives. This is particularly useful for including environmental flows (e-flows) and ecosystem services for which monetary valuation is often difficult and contested (Bingham et al., 1995; Costanza et al., 1997, 2014; Luisetti et al., 2011). The capability of MOEAs to find Pareto approximate strategies for a suite of water systems applications has been thoroughly assessed by Reed et al. (2013), and for multi-purpose* |

| Location | Comment and Response |
|---|---|
| **Reviewer 1** | |
| | *reservoir operations by Zatarain Salazar et al. (2016). In this paper, an Evolutionary Multi-Objective Direct Policy Search (EMODPS) framework is applied to map the states of a system, in this case, reservoir levels and time of the year, to actions, the release of water for different water uses (Giuliani et al., 2016; Zatarain Salazar et al., 2017). This approach has been applied to find Pareto approximate operating policies for multi-objective, multi-reservoir systems (Quinn et al., 2017; Wild et al., 2019). The motivation to use EMODPS was informed by the fact that for the selected case study, multi-objective reservoir operating policies had to be found under uncertainty. Traditional approaches for optimal control, such as stochastic dynamic programming, do not permit finding the Pareto approximate policies across multiple objectives in a single run, requiring instead that the Pareto front is constructed by testing different weights for each of the system's objectives. Such a method increases the computational burden and yields a sparse Pareto front thereby potentially missing regions of suitable policies. The use of EMODPS overcomes this challenge by generating the trade-offs across all the system's objectives simultaneously in a single algorithmic run, creating a diverse and more accurate Pareto front (Giuliani et al, 2016). This motivates the use of direct policy search, in which radial basis functions are used to find a flexible shape to map storage levels and time to release decisions for multiple objectives.* |
| Lines 92 to 93. | **Comment 1.3:**
 "As such, the implications of the trade-off on power delivery, energy prices and carbon emissions are not investigated". Dams and hydropower plants are not isolated infrastructures. The state-of-the-art is moving to evaluate the multisector implications of human-nature resources systems. Any reduction in hydropower generation not only impacts "the water demands" for this use, its impacts the power system (emission, energy prices, country economy, etc.). I recommend improving the discussion on the real implications of reducing hydropower generation in the country.

**Response:**
The main contributions of this paper are twofold. First, it explores the room for compromise in the Lower Volta by the quantifying the Pareto optimal trade-offs when e-flows previously prescribed for the basin are implemented. Secondly this paper is a new application of the EMODPS to a data scarce region where only the system goals and direction of preference are specified in the multi-objective evolutionary optimization. The manuscript therefore does not seek to evaluate the full system trade-offs comprehensively. Rather it explores the trade-offs inherent to e-flow implementation and uses an established method (EMODPS) in a new situation in doing this. So, the paper's aim lies in between a methodological advancement and evaluating system trade-offs
The authors agree that additional research is needed to assess the system trade-offs comprehensively i.e.: the real implications of meeting e-flow demands on the emissions, |

| Location | Comment and Response |
|---|---|
| **Reviewer 1** | |
| | energy prices and the economy of both Ghana and neighbouring countries to which power from the Akosombo dam is sold. This will require follow-up studies encompassing a review of the energy sector in Ghana and the politics that drive it, stakeholder engagement and feasibility studies. This is beyond the scope of the present research.

In the discussion, we touch on the potential implications of e-flow implementation when we state that:

*For instance, an increase in irrigation demand will trade-off against hydropower production at Kpong Dam and an increase in the firm energy requirement or the continuation of the de-facto policy of hydropower maximisation at Akosombo Dam, despite the availability of alternative power generation sources (Dye, 2020; Kumi, 2017), will weaken the potential for re-operation of the dam for the riverine environment. Changes in upstream water consumption as well as the construction of new dams such as the Pwalugu Dam in northern Ghana will also affect inflows to the Akosombo Dam. Gonzalez et al. (2021), however, show that practical coordination of the operation of major infrastructure in the Volta Basin, as compared to the current approach whereby dam operators fail to consider downstream built infrastructure, reduces the impact on inflows to the Akosombo Dam in particular, and also maximises basin-wide benefits. Undoubtedly this coordination should extend beyond the Volta Basin to include the entire electricity generation portfolio of Ghana and neighbouring countries to further reduce the impact of e-flows implementation in the Lower Volta on power supply.*

*In the potential re-operation of the Akosombo and Kpong dams, one has to consider that the majority of the alternative sources of power in Ghana use carbon fuels (Dye, 2020) and thus most likely contribute more to climate change compared to power generation from these two existing dams (dos Santos et al., 2006; Barros et al., 2011). It is therefore recommended that future studies encompass an overview of the energy landscape of Ghana and investigate carbon emissions, as well as examining energy price and economic implications. By exploring the room for compromise in the Lower Volta with respect to e-flows implementation this research has taken a first step towards a comprehensive assessment of the trade-offs involved at a national and local level. The potential re-operation of the Akosombo and Kpong dams can also benefit from (i) the groundwork laid by research on the pre- and post-dam river system (Lawson, 1972; Tsikata, 2008; De-Graft Johnson, 1999; Nyekodzi et al., 2018; Obirikorang et al., 2013; Adjei-Boateng et al., 2012; Owusu et al., 2022b), (ii) insights deriving from interviews and extensive stakeholder engagement (Ayivor and Ofori, 2017; Ohemeng et al., 2017; Nukpezah et al., 2017), and (iii) existing supporting legislation for e-flows implementation (L.I. 1692 Water Use Regulations, Ghana, 2001). Indeed, research on successful and stalled cases of dam re-operation indicates that stakeholder engagement and supporting legislation enhance the chances of successful e-flows implementation (Owusu et al., 2022a, 2021).*

Dye, B. J. (2020). *Structural reform and the politics of electricity crises in Ghana: tidying whilst the house is on fire?* (013 FutureDAMS Working Paper; FutureDAMS Working Paper). |

| Location | Comment and Response |
|---|---|
| **Reviewer 1** | |
| | The Discussion now also covers the application of EMODPS to the study: |
| | *Finally, the successful application of the EMODPS framework in exploring trade-offs inherent to e-flows implementation in a heavily modified river under uncertainty holds promise for similar applications elsewhere. In order to find a policy for multiple objectives in such cases, a flexible structure to map states to actions is needed.  With traditional control optimization techniques, the uncertainties need to be modelled explicitly, which creates a high computational burden and limits the ability to evaluate a large set of uncertainties (Giuliani et al., 2016). EMODPS overcomes this challenge by directly conditioning the decisions to exogenous information without requiring an explicit probabilistic model. With EMODPS, only the goals and direction of preference are required in setting up the multi-objective decision problem, making the use of this method feasible even in data scarce conditions. This study therefore concurs with Herman et al. (2020) who argue that direct policy search methods are a promising technique to enable adaptivity in water resources assessment by allowing the flexible integration of new information about the system into management decision making.* |
| | Comment 1.4: |
| | Lines 157 to 165. Is it unclear why the model was calibrated for three years with specific conditions to the system (wet, dry and normal) if the authors have access to historical data from 1981 to 2012? Why do no to calibrate the model for the full historical time-series? The calibration process produced only one reservoir operating policy for the wet, dry and normal conditions of the system, or it produces a policy by condition. |
| | Response: |
| | This was a practical decision to expedite the calibration phase of the study. The choice was therefore made to check the performance of the model against specific conditions (wet, dry and normal) using the current baseline dam operation objectives of meeting firm energy and irrigation demands and preventing flooding. In a follow-up study, we explore the vulnerabilities to hydro-climatic uncertainty (by not only using the entire historical record but also expanding upon it via synthetic hydrology to have a larger probability of sampling floods and droughts). In the case of this study, the priority is to generate the trade-offs under very little, but targeted assumptions about the hydrology of the system; in essence applying the method in a data poor situation. The reason behind the choice to calibrate for specific conditions is now stated in the manuscript: |
| | *The choice to calibrate for years with specific conditions as against the full historical time series was a practical one to expedite the calibration phase of the study.* |
| Line 116 | **Comment 1.5:** |
| | "to a steady flow of about 1,000 m3/s all year round". Is this steady flow the average flow downstream of Akosombo? Could the authors inform about the Akosombo release seasonality and monthly flow variability? |

| Location | Comment and Response |
|---|---|
| **Reviewer 1** | |
| | **Response:**
No account is taken of seasonality under the current release at Akosombo. For clarity, the text (Line 137) now reads:
*to a steady flow of about 1,000 m³/s per month all year round with no account taken of seasonality (Ntiamoa-Baidu et al., 2017).* |
| Lines 224 to 225 | **Comment 1.6:**
"While the annual firm power requirement from Akosombo Dam is 4415 GWh/year." [GWh] units refer to energy while [GW] units to power. Please, check if you are referring to "firm energy generation".

**Response:**
The authors are grateful to the reviewer for drawing our attention to this. This has been corrected to "firm energy" throughout the text. |
| | **Overall summary:**
**Part 1**: In terms of the manuscript's contribution is not clear if the authors are presenting a methodological advance or presenting the system trade-offs among different water users in the Lower Volta River basin. If the contribution is methodological, it is not clear how what the author present is different from other previous studies that use EMODPS. If the contribution it is to presents the system trade-offs, the discussion/implications of reducing hydropower generation lack broader consequences for the country and its economy. How could the results be presented to relevant decision-makers in the country if only local 'benefits' (lines 444 to 452) are given to 'validate' the implementation of hydropower reduction to support e-flow implementations? Does Ghana have alternatives to replace cheap and flexible hydropower generation? If the reduction in hydropower is replaced by thermal generation, how will this impact nationally determined contributions (in terms of $CO_2$ emission reduction)? What is the local vs country-wide trade-offs of reducing hydropower generation? I consider these questions need to be answered if the paper aims to be a viable input to any discussion on re-operating the Akosombo and Kpong hydropower plants. I recommend improving the paper's discussion and implications by including a clear discussion of the previously highlighted points.

**Response:**
The manuscript neither presents a methodological advancement nor presents an evaluation of system trade-offs. Instead, its aim is twofold, lying in between the aforementioned goals: First, it explores the room for compromise in the Lower Volta by the quantifying the Pareto optimal trade-offs when e-flows previously prescribed for the basin are implemented. Secondly this paper is a new application of the EMODPS to a data scarce region under high |

| Location | Comment and Response |
|---|---|
| **Reviewer 1** | |

uncertainty where only the system goals and direction of preference are specified in the multi-objective decision problem.

In the discussion, we touch on the potential implications of dam re-operation. The authors agree that additional research is needed to comprehensively determine the system trade-offs i.e.: the real implications of meeting e-flow demands on the emissions, energy prices and the economy of both Ghana and neighbouring countries to which power from the Akosombo dam is sold. This will require follow-up studies encompassing a review of the energy sector in Ghana and the politics that drive it, stakeholder engagement and feasibility studies. This is beyond the scope of the present research; however, our research represents a necessary first step in an exploration of the potential for e-flows implementation.

**Overall summary:**

**Part 2:** Does Ghana have alternatives to replace cheap and flexible hydropower generation? If the reduction in hydropower is replaced by thermal generation, how will this impact nationally determined contributions (in terms of CO2 emission reduction)? What is the local vs country-wide trade-offs of reducing hydropower generation? I consider these questions need to be answered if the paper aims to be a viable input to any discussion on re-operating the Akosombo and Kpong hydropower plants. I recommend improving the paper's discussion and implications by including a clear discussion of the previously highlighted points.

**Response:**

The authors agree that additional research is needed for the results to be used by relevant decision makers. This will require comprehensive follow-up studies encompassing an overview of the energy landscape of Ghana, stakeholder engagement and feasibility studies on energy price, carbon emissions, etc. This is beyond the scope of the current research; however, this research is a necessary first step in exploring trade-offs inherent to e-flows implementation.

In response to the comment on alternatives to hydropower, we would like to highlight that Ghana does have alternatives to hydropower. These alternatives are not necessarily cheap. Agreements were made that Ghana must pay 90% of the cost of energy from these alternative sources irrespective of whether they are used or not. This provides an incentive to use the alternative sources. As Dye (2020, Page 4) says in his overview of the energy situation in Ghana and the politics that drive it, "this crisis of shortage was quickly replaced with one of overabundance: Ghana went into a power plant construction overdrive, resulting in electricity-generation capacity equalling twice the country's demand by 2018". He goes further explaining that: "This increase is particularly problematic as it came from 'take-or-pay' contracts that involve the government's distribution utility, the Electricity Company of Ghana (ECG), promising to pay private electricity companies typically for 90% of the power they make available, regardless of whether it is used. Ghana's large imbalance in supply and demand is leaving a costly bill, reaching 4%–5% of GDP in 2018 (World Bank, 2018)".

| Location | Comment and Response |
|---|---|
| **Reviewer 1** | |
| | Dye, B. J. (2020). *Structural reform and the politics of electricity crises in Ghana: tidying whilst the house is on fire?* (013 FutureDAMS Working Paper; FutureDAMS Working Paper). |

The contribution of the present study has been clarified in the introduction:

*The main contribution of the paper is twofold: First, it explores the room for compromise in the Lower Volta by the quantifying the Pareto approximate trade-offs when e-flows previously prescribed for the basin are implemented. Secondly this paper is a new application of the EMODPS to a data scarce region under high uncertainty where only the system goals and direction of preference are specified in the multi-objective decision problem.*

In Discussion, we touch on the potential implications of e-flows implementation:

*For instance, an increase in irrigation demand will trade-off against hydropower production at Kpong Dam and an increase in the firm energy requirement or the continuation of the de-facto policy of hydropower maximisation at Akosombo Dam, despite the availability of alternative power generation sources (Dye, 2020; Kumi, 2017), will weaken the potential for re-operation of the dam for the riverine environment. Changes in upstream water consumption as well as the construction of new dams such as the Pwalugu Dam in northern Ghana will also affect inflows to the Akosombo Dam. Gonzalez et al. (2021), however, show that practical coordination of the operation of major infrastructure in the Volta Basin, as compared to the current approach whereby dam operators fail to consider downstream built infrastructure, reduces the impact on inflows to the Akosombo Dam in particular, and also maximises basin-wide benefits. Undoubtedly this coordination should extend beyond the Volta Basin to include the entire electricity generation portfolio of Ghana and neighbouring countries to further reduce the impact of e-flows implementation in the Lower Volta on power supply.*

*In the potential re-operation of the Akosombo and Kpong dams, one has to consider that the majority of the alternative sources of power in Ghana use carbon fuels (Dye, 2020) and thus most likely contribute more to climate change compared to power generation from these two existing dams (dos Santos et al., 2006; Barros et al., 2011). It is therefore recommended that future studies encompass an overview of the energy landscape of Ghana and investigate carbon emissions, as well as examining energy price and economic implications. By exploring the room for compromise in the Lower Volta with respect to e-flows implementation this research has taken a first step towards a comprehensive assessment of the trade-offs involved at a national and local level. The potential re-operation of the Akosombo and Kpong dams can also benefit from (i) the groundwork laid by research on the pre- and post-dam river system (Lawson, 1972; Tsikata, 2008; De-Graft Johnson, 1999; Nyekodzi et al., 2018; Obirikorang et al., 2013; Adjei-Boateng et al., 2012; Owusu et al., 2022b), (ii) insights deriving from interviews and extensive stakeholder engagement (Ayivor and Ofori, 2017; Ohemeng et al., 2017; Nukpezah et al., 2017), and (iii) existing supporting legislation for e-flows implementation (L.I. 1692 Water Use Regulations, Ghana, 2001). Indeed, research on successful and stalled cases*

| Location | Comment and Response |
|---|---|
| **Reviewer 1** | |
| | *of dam re-operation indicates that stakeholder engagement and supporting legislation enhance the chances of successful e-flows implementation (Owusu et al., 2022a, 2021).* |
| | And also discuss the application of EMODPS: |
| | *Finally, the successful application of the EMODPS framework in exploring trade-offs inherent to e-flows implementation in a heavily modified river under uncertainty holds promise for similar applications elsewhere. In order to find a policy for multiple objectives in such cases, a flexible structure to map states to actions is needed. With traditional control optimization techniques, the uncertainties need to be modelled explicitly, which creates a high computational burden and limits the ability to evaluate a large set of uncertainties (Giuliani et al., 2016). EMODPS overcomes this challenge by directly conditioning the decisions to exogenous information without requiring an explicit probabilistic model. With EMODPS, only the goals and direction of preference are required in setting up the multi-objective decision problem, making the use of this method feasible even in data scarce conditions. This study therefore concurs with Herman et al. (2020) who argue that direct policy search methods are a promising technique to enable adaptivity in water resources assessment by allowing the flexible integration of new information about the system into management decision making.* |
| | **Overall summary:** |
| | **Part 3:** It is not clear how robust are the presented results in terms of future climate change impacts or upstream water uses changes. The Akosombo and Kpong hydropower plants are located downstream of the Volta River basin, and any change in water availability upstream in the basin could possibly impact the system evaluated in this study. Methods such as Sensitivity Analysis or Exploratory modelling evaluating 'a wider range' of future states of the system could provide more conclusive results and improve the discussion of future impacts of changes in water availability. |
| | **Response:** |
| | As a follow up to this present study and as part of an MSc thesis (Buskop, 2022), an uncertainty evaluation has been carried out for the optimised release policies from this study for 2400 scenarios representing different states of the world selected through the Latin Hypercupe sampling. The robustness for each policy is compared across different robustness metrics (Buskop, 2022). The main results are in line with the current study in that: |
| | "Using various robustness metrics to define stakeholder preferences across objectives shows the same trade-offs as in the unmodified objective scores. Environmental policies do not perform well with energy objectives, while the hydropower policies do not work well for the irrigation and environmental objectives" |

| Location | Comment and Response |
|---|---|
| **Reviewer 1** | |
| | On changes to upstream water use, Buskop (2022) states: "When looking into the importance of included uncertainties and system levers, it was seen that the water usages of Togo and Côte d'Ivoire play a significant role in the system."

 The findings from this MSc thesis are being prepared for journal publication.

 Buskop, T. (2022). *Will the Benefits Keep Flowing?* MSc thesis, TU Delft. Available at: https://repository.tudelft.nl/islandora/object/uuid%3A02d941cf-b7a8-4498-9a13-762c7e6988fc?collection=education |
| **Reviewer 2** | |
| Lines 365-367: | **Comment 2.1:**
 The authors seem to suggest that the flood control objective is performing less well for e-flows2 and 3 (0.83 down from 0.99 in clam e-flows. This is however not shown in Figure 4. Instead Figure 4 suggests the performance of the flood control objective remains the same across all e-flows.

 **Response:**
 The authors are grateful to the reviewer for highlighting the ambiguity in the text. The performance of the flood objective does remain the same, but the performance of the solution decreases for the environment objective for e-flows 2 and 3.
 The text has been updated to read:
 *The solutions for all e-flow configurations perform well for the flood control objective even though e-flow 2 and 3 prescribe flood releases for two months of the year. As such, comparing clam e-flows to e-flows 2 and 3, there is a reduction (0.99 for clam e-flows vs. 0.83 for e-flows 2 and 3) in the performance of the 'best ==environment'== solution for the latter two, as expected, showing that the requirement for floods for two months in a year in those e-flow configurations are not met*. |

---

## Author Response (AR2)

**Quantifying the trade-offs in re-operating dams for the environment in the Lower Volta River**

Afua Owusu[a b], Jazmin Zatarain Salazar[b], Marloes Mul[a], Pieter van der Zaag[a c], Jill Slinger[b d]

[a] Land and Water Management Department, IHE Delft Institute for Water Education, Westvest 7, 2611 AX Delft, The Netherlands
[b] Faculty of Technology, Policy and Management, TU Delft, Jaffalaan 5, 2628 BX Delft, The Netherlands
[c] Faculty of Civil Engineering and Geosciences, TU Delft, Stevinweg 1, 2628 CN Delft, The Netherlands
[d] Institute of Water Research, Rhodes University, Drosty Rd, Grahamstown, 6139, South Africa

**Response to reviewers:**
The authors thank the reviewer and editor for their comment on expanding the discussion on the impact of hydropower reduction on the Ghanaian economy and power system operation costs. Below is the detailed response to this comment.

| Location | Comment and Response |
|---|---|
| Discussion | **Comment:**
The reviewer has a minor comment regarding your discussion and suggest you expand on how hydropower reduction can impact Ghanaian economy and power system operation costs. I look forward to receiving your revised manuscript.

**Response:**
The authors are grateful for this comment and have updated the Discussion to include a paragraph on the potential impact of dam re-operation on the energy landscape of Ghana, carbon emissions from the country, as well as energy pricing and economic implications. In lines 457 to 481, we write:

*Expanding on the current electricity generation portfolio of Ghana, the contribution of other renewable energy sources besides hydropower to the power mix has remained under 1% since 2000, despite an on-grid target of 10% by 2020 (now extended to 2030) (Acheampong et al., 2021; Energy Commission Ghana, 2022). The alternative sources of electricity in Ghana use carbon fuels for thermal power generation, accounting for approximately 65% of the electricity generation portfolio in 2020 (Dye, 2020; Acheampong et al., 2021; Energy Commission Ghana, 2022). It is expected that these alternative carbon-based power sources contribute more to climate change compared to power generation from Akosombo and Kpong dams due to the fact greenhouse gas emissions from hydropower dams is negatively correlated with dam age and even the more recent dam, Kpong, has been in operation for over 35 years (dos Santos et al., 2006; Barros et al., 2011). As such, dam re-operation in Ghana may have the long-term environmental and economic consequences of higher greenhouse gases emissions if it results in a higher reliance on the existing carbon-based power generation options, rather than other renewables like solar and wind power. Furthermore, in Ghana, hydropower has traditionally been a cheaper source of electricity compared to fossil fuel-based power generation and as* |

| Location | Comment and Response |
|---|---|
|  | *Ghana has increased its reliance on the latter, electricity generation costs have increased resulting in higher tariffs for consumers (Energy Commission Ghana, 2022; Public Utilities Regulatory Commission, 2015). Finally, any reduction in hydropower production from Akosombo and Kpong dams due to re-operation may result in reduced overall electricity supply in Ghana as experienced during periods of drought in the past (Dye, 2020). It is estimated that the negative economic impacts of power shortages and load shedding, such as decreased productivity in industries, loss of revenue for businesses, and increased costs for backup power sources led to a GDP reduction of about 1.8-2% during the 2014-2016 power crisis (Acheampong et al., 2021). Considering these potential adverse impacts of dam re-operation, it is recommended that future studies encompass a deeper analysis of the energy landscape of Ghana and investigate carbon emissions and the path to greener energy in the country, as well as energy pricing and economic implications.*

Acheampong, T., Menyeh, B. O., and Agbevivi, D. E.: Ghana's Changing Electricity Supply Mix and Tariff Pricing Regime: Implications for the Energy Trilemma, Oil, Gas & Energy Law, 19, 2021.
Barros, N., Cole, J. J., Tranvik, L. J., Prairie, Y. T., Bastviken, D., Huszar, V. L. M., Del Giorgio, P., and Roland, F.: Carbon emission from hydroelectric reservoirs linked to reservoir age and latitude, Nat Geosci, 4, 593–596, https://doi.org/10.1038/ngeo1211, 2011.
Dye, B. J.: Structural reform and the politics of electricity crises in Ghana: tidying whilst the house is on fire?, Manchester, 2020.
Energy Commission Ghana: 2022 National Energy Statistics, Accra, 2022.
Public Utilities Regulatory Commission: Public Utilities Regulatory Commission Press Release: Approved Electricity and Water Tariffs Effective 14th December 2015 , 2015.
dos Santos, M. A., Rosa, L. P., Sikar, B., Sikar, E., and dos Santos, E. O.: Gross greenhouse gas fluxes from hydro-power reservoir compared to thermo-power plants, Energy Policy, 34, 481–488, https://doi.org/10.1016/J.ENPOL.2004.06.015, 2006. |

---

## Author Response (AR3)

**Quantifying the trade-offs in re-operating dams for the environment in the Lower Volta River**

Afua Owusu[a][b], Jazmin Zatarain Salazar[b], Marloes Mul[a], Pieter van der Zaag[a][c], Jill Slinger[b][d]

[a] Land and Water Management Department, IHE Delft Institute for Water Education, Westvest 7, 2611 AX Delft, The Netherlands
[b] Faculty of Technology, Policy and Management, TU Delft, Jaffalaan 5, 2628 BX Delft, The Netherlands
[c] Faculty of Civil Engineering and Geosciences, TU Delft, Stevinweg 1, 2628 CN Delft, The Netherlands
[d] Institute of Water Research, Rhodes University, Drosty Rd, Grahamstown, 6139, South Africa

**Response to reviewers:**
The authors thank the editor for their comments on improving the clarity of the abstract and the data availability statement. Below are the changes made in response to the comments.

| Location | Comment and Response |
|---|---|
| Discussion | **Comment:** |
| | Thank you for your revised manuscript. You have addressed the comments and I am happy to accept your manuscript for publication subject to minor revisions (review by editor). I list the corrections as follows. |
| Line 13 | Line 13 Ghana as a whole, has enjoyed vast economic benefits |
| | Change to "Ghana has enjoyed vast economic benefits" |
| Line 24-28: | Line 24-28: It is found that climate change leading to increased annual inflows to the Akosombo Dam reduces the trade-off between hydropower and the environment while climate change resulting in lower inflows provides the opportunity to strategically provide dry season environmental flows, that is, reduce flows sufficiently to meet low flow requirements for key ecosystem services such as the clam fishery. |
| | You said 'leading to increased annual inflows to the Akosombo Dam', then later stated 'while climate change resulting in lower inflows'. This is confusing. Please correct. Please also break this sentence down into shorter sentences. |
| Line 532-534 | Line 532-534 Data availability |
| | Please update this for the final published version. Please also specify what data will be made available and how the data can be accessed. |
| | **Response:** |
| | The authors thank the editor for accepting the paper for publication and have made the following changes as requested: |

| Location | Comment and Response |
|---|---|
| | Line 13: *In contrast to the costs borne by those in the vicinity of the river, Ghana has enjoyed vast economic benefits from the affordable hydropower, irrigation schemes and lake tourism that developed after construction of the dams.*

Line 24-28: *It is found that climate change leading to increased annual inflows to the Akosombo Dam reduces the trade-off between hydropower and the environment as this scenario makes more water available for users. Furthermore, climate change resulting in lower annual inflows provides the opportunity to strategically provide dry season environmental flows, that is, reduce flows sufficiently to meet low flow requirements for key ecosystem services such as the clam fishery.*

Line 532-534: The data availability statement now reads:
*The hydrological and hydraulic data associated with this manuscript, specifically, historical water levels, dam releases, and storage-area equations for the Akosombo and Kpong dams, will be made available upon consultation with the national organisation, Volta River Authority (VRA), that owns the data. Requests for these data may be made to the corresponding author.*

*The model code for the running the Evolutionary Multi-Objective Direct Policy Search for the Akosombo and Kpong dams is publicly available on Github at: https://github.com/Afua-O/Vol_Opt.git* |

---

## Author Response (AR4)

**Quantifying the trade-offs in re-operating dams for the environment in the Lower Volta River**

Afua Owusu[a b], Jazmin Zatarain Salazar[b], Marloes Mul[a], Pieter van der Zaag[a c], Jill Slinger[b d]

[a] Land and Water Management Department, IHE Delft Institute for Water Education, Westvest 7, 2611 AX Delft, The Netherlands
[b] Faculty of Technology, Policy and Management, TU Delft, Jaffalaan 5, 2628 BX Delft, The Netherlands
[c] Faculty of Civil Engineering and Geosciences, TU Delft, Stevinweg 1, 2628 CN Delft, The Netherlands
[d] Institute of Water Research, Rhodes University, Drosty Rd, Grahamstown, 6139, South Africa

**Response to reviewers:**
The authors thank the editor for their comment and hope the explanation below makes these statements clear.

| Location | Comment and Response |
|---|---|
|  | **Comment:**
One final issue for Line 24-28: You said 'leading to increased annual inflows to the Akosombo Dam', then later stated 'while climate change resulting in lower inflows'. This is confusing. Please explain.

**Response:**
The two statements refer to 2 situations: one where there is increased inflows and the other where there is decreased inflows. We have changed the wording in '*while climate change resulting in lower inflows*' from "*lower*" to "*decreased*" to make the contrast clearer.

Based on a literature review (Section 3.2) we designed 5 scenarios indicative of the range of climate-induced changes predicted for the Volta discharge for the mid to long term. These are listed in Table 1 and the effects of different climate futures on the Pareto approximate solutions for the Volta basin are presented in Figure 5 and section 4.2.
In the abstract when we state: "*It is found that climate change **leading to increased annual inflows** to the Akosombo Dam reduces the trade-off between hydropower and the environment as this scenario makes **more water available for users**.*" we are referring to the scenarios that lead to an increase in annual inflows. We find that in these scenarios, there is more water compared to the current baseline so that the trade-off between water users is reduced.
The second line: "*Furthermore, climate change resulting in ==decreased== annual inflows provides the opportunity to **strategically provide dry season environmental flows**, that is, reduce flows sufficiently **to meet low flow requirements for key ecosystem services** such as the clam fishery*" refers to the scenario which leads to a decrease in annual inflows to the Akosombo dam. In this scenario, while there is less water for all users, it is still possible to reap some environmental benefits out of a 'bad' situation because it is possible to strategically release recommended dry season e-flows. In effect, if there is less water, there will be low releases from Akosombo Dam and the annual firm energy demand cannot be met, but some |

| Location | Comment and Response |
|---|---|
| | ecosystems like the clam fishery require these low flows at certain times of the year and in this scenario, we can meet this requirement without trading off against hydropower because we cannot meet the hydropower demand anyway (in Figure 5, this is most clearly illustrated in scenario 1-clam e-flows graph, where the best pareto solutions for the environment *is not* significantly different from the best hydropower solution on the Kpong (Kp) and Akosombo (Ak) axes). |
| | In the Discussion (Lines 426 to 443), an explanation of the two statements in the Abstract can be found:
*"While the majority of climate predictions for the Volta River generally point to an increase in annual water availability (Kunstmann and Jung, 2005; Aerts et al., 2006; Jung et al., 2012; Abubakari, 2021; Jin et al., 2018; Sylla et al., 2018),* **based on this study, an argument can be made that both an increase or a decrease in inflows to the Lower Volta enhance the potential for e-flows implementation compared to the current baseline.** *On the one hand, an increase in inflows to the Akosombo dam as applied in scenario 3, reduces the amount of the firm energy requirement that would have to be supplemented by other sources for the implementation of 'fair environmental solutions' to about 3.9% (vs 11.6%) for clam e-flows, and then 23.2% (vs 32%) for e-flows 2 and 33.7% (vs 38%) for e-flows 3. On the other hand, a decrease in inflows to the Akosombo Dam, whereby at best only 2,774 GWh/year of hydropower can be generated, provides opportunity to strategically release recommended dry season e-flows to reap some environmental benefits out of a 'bad' situation where annual flow releases from the dam will be low anyway. This operation policy under dry climate scenarios could also be adopted in dry years, in essence modelling the Episodic E-flows Implementation approach, which is an opportunistic approach to dam re-operation that takes advantage of prevailing hydrological conditions (Warner et al., 2014; Yang and Yang, 2014; Owusu et al., 2021). This contrasts with the alternative approaches, Adaptive Management and Blanket Operation which represent more structural inclusion of e-flows in the dam operation policy (Warner et al., 2014)."*

To elaborate on the requirement for low/dry season flows and why this is important for the clam fishery: this is because at certain times of the year (Nov to March), in the veliger larva life stage of the Volta clam, low water levels encourage salt intrusion to the Lower Volta and while adult clams are freshwater species with some tolerance for short term saline conditions, the larva of the clam require salinity.
In Owusu *et al.*, 2022, more details on the Volta clam, their lifecycle and habitat and the derivation of this environmental flow (clam e-flows) requirement can be found.

Owusu, A., Mul, M., Strauch, M., van der Zaag, P., Volk, M., and Slinger, J.: The clam and the dam: A Bayesian belief network approach to environmental flow assessment in a data scarce region, Science of The Total Environment, 810, 151315, https://doi.org/10.1016/J.SCITOTENV.2021.151315, 2022. |

---

## Author Response (AR5)

The type of lines have been changed in Figure 3 in the manuscript and in Figures S1 to S3 in the supplementary material.

For figures 4 and 5, these are parallel axis graphs each showing between 200 to 400 lines ('optimal' solutions), falling into six different groupings, and crossing five axes. Unfortunately, using different types of dashed lines does not work as the 200+ lines on each graph are clustered together, and intersecting at multiple points in each graph. Since only black and grey (on a white background) are clearly discernable to all the different types of colour vision deficiencies, there are not enough colour options to show the six groupings for the 200+ lines on each graph.

Is it possible to add a note in the captions referring to the table below which explains how the colours change for the different types of colour blindness?

This offers a solution for all except Monochromacy. Unfortunately, we have so far been unable to think up a way to present these graphs clearly for those with monochromacy and would be grateful for advice on how to do this.

| Type of solution | Normal | Red Weak | Red blind | Green Weak | Green blind | Blue blind |
|---|---|---|---|---|---|---|
| Best hydropower | Red | Red | Dark grey | Red | Light brown | Red |
| Fair hydropower | Light red/pink | Light blue | Light blue | Light blue | Light blue | Light red/pink |
| Best environment | Green | Green | Dark yellow/Brown | Green | Brown | Green |
| Fair environment | Light green | Yellow/light green | Yellow | Light green | Yellow | Light green |
| Best irrigation | Purple | Blue | Blue | Blue | Blue | Dark grey |
| Other solutions | Light grey | Light grey | Light grey | Light orange | Light orange | Light grey |

NB: For Blue-Weak and Blue-Cone monochromancy, the colours remain unchanged from Normal.